# Phase transition in the cuprates from a magnetic-field-free stiffness meter viewpoint

Itzik Kapon[1], Zaher Salman [2], Itay Mangel[1], Thomas Prokscha [2], Nir Gavish[3] & Amit Keren [1]

A method to measure the superconducting (SC) stiffness tensor $\bar{\rho}_s$, without subjecting the sample to external magnetic field, is applied to $La_{1.875}Sr_{0.125}CuO_4$. The method is based on the London equation $\mathbf{J} = -\bar{\rho}_s\mathbf{A}$, where $\mathbf{J}$ is the current density and $\mathbf{A}$ is the vector potential which is applied in the SC state. Using rotor free $\mathbf{A}$ and measuring $\mathbf{J}$ via the magnetic moment of superconducting rings, $\bar{\rho}_s$ at $T \rightarrow T_c$ is extracted. The technique is sensitive to very small stiffnesses (penetration depths on the order of a few millimeters). The method is applied to two different rings: one with the current running only in the $CuO_2$ planes, and another where the current must cross planes. We find different transition temperatures for the two rings, namely, there is a temperature range with two-dimensional stiffness. Additional low energy muon spin rotation measurements on the same sample determine the stiffness anisotropy at $T < T_c$.

[1] Department of Physics, Technion—Israel Institute of Technology, 3200003 Haifa, Israel. [2] Laboratory for Muon Spin Spectroscopy, Paul Scherrer Institute, 5232 Villigen PSI, Switzerland. [3] Department of Mathematics, Technion—Israel Institute of Technology, 3200003 Haifa, Israel. Correspondence and requests for materials should be addressed to I.K. (email: itzikapon@gmail.com) or to A.K. (email: keren@physics.technion.ac.il)

The existence of two-dimensional (2D) superconductivity (SC) in the $CuO_2$ planes of the cuprates has been demonstrated by either isolated $CuO_2$ sheets[1,2], or in bulk, by applying a magnetic field parallel to these planes[3–5]. In the vicinity of charge stripes formation, the layers are so well decoupled[6] that, in fact, two transition temperatures have been found by resistivity[7] and magnetization in needle-shaped samples[8], where the demagnetization factor tends to zero, and the measured susceptibility equals the intrinsic one. The magnetization measurements were done in both $c$-needles, where the $CuO_2$ planes are perpendicular to the field direction, and $a$-needles where the planes are parallel to the field. An updated phase diagram showing the magnetization critical temperature in $c$-needles $T_M^c$ and $a$-needles $T_M^a$ is presented in Fig. 1. The resistivity critical temperature $T_\rho^c$ of the same samples agrees with $T_M^a$. The inset shows an example of such magnetization measurement for $La_{2-x}Sr_xCuO_4$ (LSCO) with $x = 0.12$.

However, zero resistivity and diamagnetism do not require bulk superconductivity and can occur due to superconducting islands or filaments. It is not clear whether the observed in-plane superconductivity is a macroscopic phenomenon and if the sample supports global 2D stiffness as expected from Kosterlitz–Thouless–Berezinskii (KTB) theory[9–11]. If it does, there should be a temperature (and doping) range where the intra-plane stiffness $1/\lambda_{ab}^2$ is finite, while the inter-plane stiffness $1/\lambda_c^2$ is zero ($\lambda$ is the penetration depth).

Here we examine the possibility of macroscopic 2D superconductivity in the bulk using two different techniques: Low energy muon spin rotation (LE-$\mu$SR) and Stiffnessometer. The Stiffnessometer is a method we developed to measure particularly small SC stiffness. We focus on the "anomalous doping" $x = 1/8$ regime, where the difference between the two transition temperatures is large, and minute inhomogeneity of Strontium doping does not lead to significant deviations in the transition temperatures. (Details of materials preparation are given in the Methods section.) Our major finding is that in LSCO $x = 1/8$ there is a temperature interval of 0.7 K where global 2D SC stiffness exists in the planes, while it is zero between them. In this interval supercurrent can only flow in the $CuO_2$ planes.

## Results

**Stiffnessometer experimental set-up and measurements.** The Stiffnessometer is based on the fact that outside an infinitely long coil, the magnetic field is zero while the vector potential $\mathbf{A}$ is finite. When such a coil is threaded through a superconducting ring, the vector potential leads to supercurrent density $\mathbf{J}$ according to the London equation

$$\mathbf{J} = -\bar{\rho}_s \mathbf{A}, \qquad (1)$$

where $\bar{\rho}_s$ is the stiffness tensor. This current flows around the ring and generates a magnetic moment. We detect this moment by moving the ring and the inner coil (IC) rigidly relative to a gradiometer, which is a set of pickup loops wound clockwise and anticlockwise. The gradiometer is placed in the center of a bigger coil which is used to cancel stray field at the ring position. This field is measured with an open ring where supercurrent cannot close a loop; hence, any signal is due to the field only. This procedure and a discussion on the coil being finite are described in Kapon et al.[12] and reviewed in Supplementary Note 1.

The experimental set-up and our coil and ring are presented in Fig. 2a. The current generated in the gradiometer by the inner coil and the sample movement is measured by a SQUID magnetometer. The measurements are done in zero gauge-field cooling procedure, namely, the ring is cooled to a temperature below $T_c$, and only then the current in the inner coil is turned on. It is the change in magnetic flux inside the inner coil that creates an electric field in the ring, and sets persistent currents in motion.

There is a range of applied currents for which using the London equation rather than the full Ginzburg–Landau theory or Pippard relation is justified: (1) The phase of the SC order parameter, $\varphi$, obeys $\nabla\varphi = l/r$ where $l$ is an integer. Cooling at $\mathbf{A} = 0$ must set $l = 0$ to minimize the kinetic energy. This value of $l$ does not change as $\mathbf{A}$ is turned on, and London's equation holds as long as $A < A_c(T)$, where $A_c(T)$ is the critical value of the vector potential which tends to zero as $T \to T_c$. (2) The coherence length $\xi$ is much shorter than $\lambda$ for all the measured temperatures[12]. Further details are available in Supplementary Note 1.

To examine the orientation-dependent response of LSCO to different directions of $\mathbf{A}$, we cut two types of rings from a single crystal rod: a "$c$-ring" where the crystallographic $\hat{\mathbf{c}}$ direction is parallel to the ring symmetry axis, i.e. the supercurrent flows in the $CuO_2$ planes, and an "$a$-ring" where the crystallographic $\hat{\mathbf{a}}$ direction is parallel to the ring symmetry axis, i.e. the supercurrent travels both in the planes and between them. The rings, shown in Fig. 3a, have inner radius of 0.5 mm, outer radius of 1.5 mm and 1 mm height.

The inset of Fig. 2b presents raw Stiffnessometer data of $c$-ring taken with inner coil current of 1 mA. The vertical axis is the measured voltage by the SQUID. The horizontal axis is the position $z$ of the ring relative to the center of the gradiometer. The red data points are measured above $T_c$ and represent the signal generated by the inner coil alone. The blue points are measured below $T_c$ and correspond to the inner coil and the ring. The difference between them, $\Delta V_R(z)$, is the signal from the ring itself. This signal is shown in Fig. 2b for different temperatures. Between 4.5 and 27 K there is hardly any change in the signal, because the Stiffnessometer is not sensitive to short penetration depth compared to the sample size. However, above 28 K the signal drops dramatically fast with increasing temperature.

We define the peak-to-peak voltage of the rings and the inner coil, $\Delta V_R^{max}$ and $\Delta V_{IC}^{max}$ respectively, as shown in Fig. 2b. Their ratio holds the information about the stiffness, as we explain shortly. Figure 3a presents $\Delta V_R^{max}$ of both rings. These voltages are normalized by their maximal value for comparison purposes.

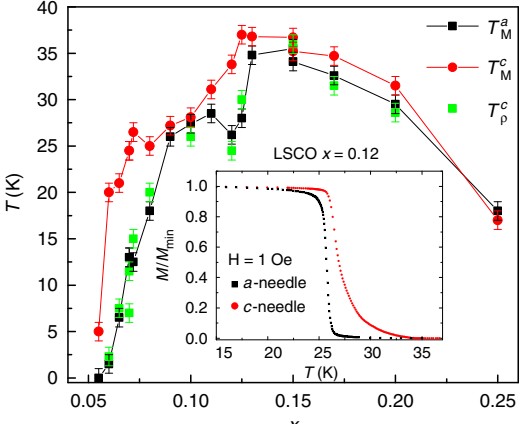

**Fig. 1** LSCO phase diagram. Temperature versus Sr doping $x$ for $a$- and $c$-needles. $T_M$ is the on-set temperature of diamagnetic signal taken from magnetization measurement and $T_\rho$ is the temperature where the resistivity goes to zero. Error bars represent the uncertainty in determining the transition temperature. The inset introduces an example of magnetization measurement for two $x = 0.12$ needles at $H = 1$ Oe

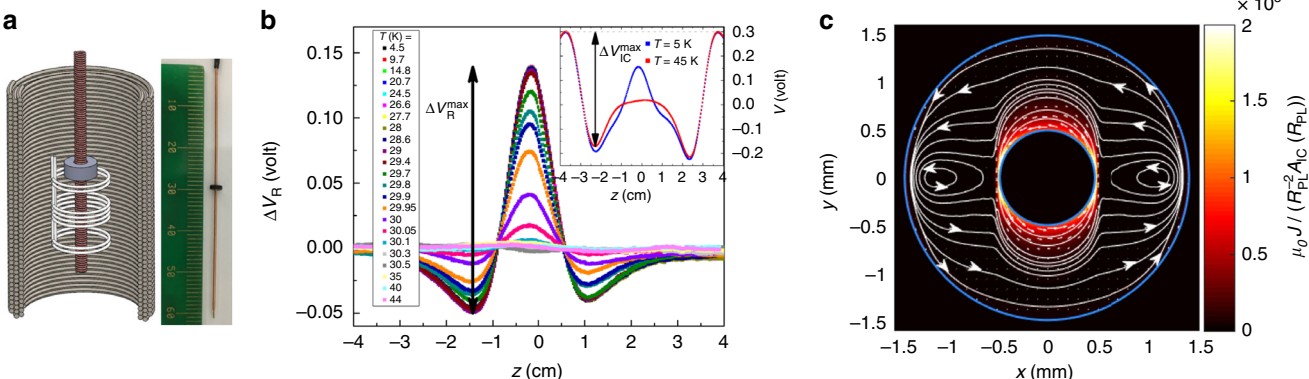

**Fig. 2** Stiffnessometer. **a** An illustration of the Stiffnessometer operation principal and a photo of typical ring and coil with 2400 windings. The long coil is threaded through a ring and they both move with respect to a gradiometer which is connected to a SQUID. The SQUID measures the flux through the gradiometer and hence the average vector potential on it $\langle A^{\theta} \rangle$. **b** Temperature dependence of an LSCO $x = 0.125$ c-ring signal as measured by the Stiffnessometer with $I = 1$ mA in the inner coil. The data presented are after subtraction of the coil contribution as explained in the text. The inset shows raw Stiffnessometer data for a temperature above and below $T_c$. The difference is due to the ring contribution. **c** The currents streamlines in a-ring at midheight ($z = 0$) derived from the solution of Eq. (3) with $\lambda_c = 145$ μm and $\lambda_{ab} = 13.9$ μm. The false colors show the current intensity. Naturally the flow is not isotropic. Vortices develop on both sides of the $x$ axis

We detect two different stiffness transition temperatures, $T_s^c = 30.1$ K for the c-ring, and a lower one $T_s^a = 29.4$ K for the a-ring. We also examine the influence of the inner coil current on the transition. Data corresponding to three different currents are shown in the figure. Below 1 mA there is no change in the transition, which otherwise widens and appears at slightly lower temperature.

The inset of Fig. 3a demonstrates a linear relation between $\Delta V_R^{max}$ and $I$ (in the inner coil) for both rings up to some critical current $I_c$. This linearity indicates the validity regime of the London equation. The measurement for the c-ring was done at constant temperature $T = 29.92(5)$ K close to $T_s^c$ and well above $T_s^c$ where the a-ring cannot support supercurrent. The temperatures up to which the London equation is valid are labeled in the main figure. The validity of the London equation is further discussed in Supplementary Note 1.

The Stiffnessometer data reveal a new phenomenon. There is a temperature range with finite 2D stiffness in the planes, although supercurrent cannot flow between them. In other words, upon cooling, the SC phase transition starts by establishing a global 2D stiffness, and only at lower temperature a true 3D superconductivity is formed.

**Stiffnessometer data analysis.** To analyze the data, we relate the measured voltage to the vector potential. Since SQUID measures flux, and the vector potential on the gradiometer is proportional to the flux threading it, the ratio of the peak-to-peak voltages satisfies

$$\frac{\Delta V_R^{max}}{\Delta V_{IC}^{max}} = G \frac{\langle A_R^{\theta}(R_{PL}) \rangle}{A_{IC}^{\theta}(R_{PL})}, \quad (2)$$

where $A_R^{\theta}$ and $A_{IC}^{\theta}$ are the rings and inner coil vector potential components in the azimuthal direction $\hat{\theta}$ respectively, $R_{PL}$ is the gradiometer radius, $\langle \rangle$ stands for averaging over the pickup loops, and $G$ is a geometrical factor determined experimentally (Supplementary Note 1).

In order to extract $\bar{\rho}_s$ from the voltages ratio of Eq. (2), we must determine the dependence of $\mathbf{A}_R(R_{PL})$ on the stiffness. This is done by solving the combined Maxwell and London's equation

$$\nabla \times \nabla \times \mathbf{A}_R = \bar{\rho}_s \left( \mathbf{A}_R + \frac{\Phi_{IC}}{2\pi r} \hat{\theta} \right), \quad (3)$$

where $\Phi_{IC}$ is the flux through the inner coil, and $\bar{\rho}_s$ is finite only inside the ring. For c-ring $\bar{\rho}_s$ is merely a scalar and equals $\lambda_{ab}^{-2}$. For a-ring, it is diagonal in Cartesian coordinates, with $\rho_{xx} = \lambda_c^{-2}$ and $\rho_{yy} = \rho_{zz} = \lambda_{ab}^{-2}$.

We solve Eq. (3) numerically for our rings geometry and various $\lambda_{ab}$ and $\lambda_c$ with FreeFEM++[13] and Comsol 5.3a. The c-ring solution, which is sensitive to $\lambda_{ab}$ only, is discussed in Kapon et al.[12]. More information on the anisotropic solution can be found in Supplementary Note 2. Using Eq. (2), the numerical solution, and the data in Fig. 2b we extract $1/\lambda_{ab}^2$, and plot it in Fig. 3b on a semi-log scale (blue solid spheres). The extraction of $\lambda_{ab}$ is from our $I = 1$ mA data and is valid only as long as $I < I_c$. This point is marked by green arrow in the inset. Beyond that point the calculated values only give qualitative trend but hardly the actual values of $\lambda_{ab}(T)$.

In order to extract $\lambda_c$ we have to know $\lambda_{ab}$ at the temperatures of interest. As can be seen from Fig. 3a the c-ring Stiffnessometer measurements are in saturation just when a-ring stiffness becomes relevant. Therefore, we applied LE-$\mu$SR to the same samples.

**LE-$\mu$SR measurements.** In LE-$\mu$SR, spin-polarized muons are injected into a sample. By controlling the muons energy $E$ between 3 and 25 keV, the muons stop with high probability at some chosen depth inside the sample while keeping their polarization intact. The stopping profile $p(x, E)$, where $x$ is stopping depth, is simulated by the TRIM.SP Monte Carlo code[14]. Figure 4 presents the stopping profiles in LSCO for different implantation energies. For each energy, we parameterize the stopping profile using

$$p(x, E) = \frac{p_0 (x_0 - x)^3}{\exp[(x_0 - x)/\zeta] - 1} H(x_0 - x). \quad (4)$$

Here $x_0$ is some cut-off position the muon cannot cross and is energy dependent, $H(x_0 - x)$ is Heaviside's function, and $\zeta$ and $p_0$ are energy-dependent free parameters. The energy dependence of the fit parameters is given in the Methods section.

When an external magnetic field is applied, the muon spin precesses at the Larmor frequency corresponding to the field. We assume an exponential decay of the magnetic field with a characteristic length of $\lambda$ along the direction perpendicular to

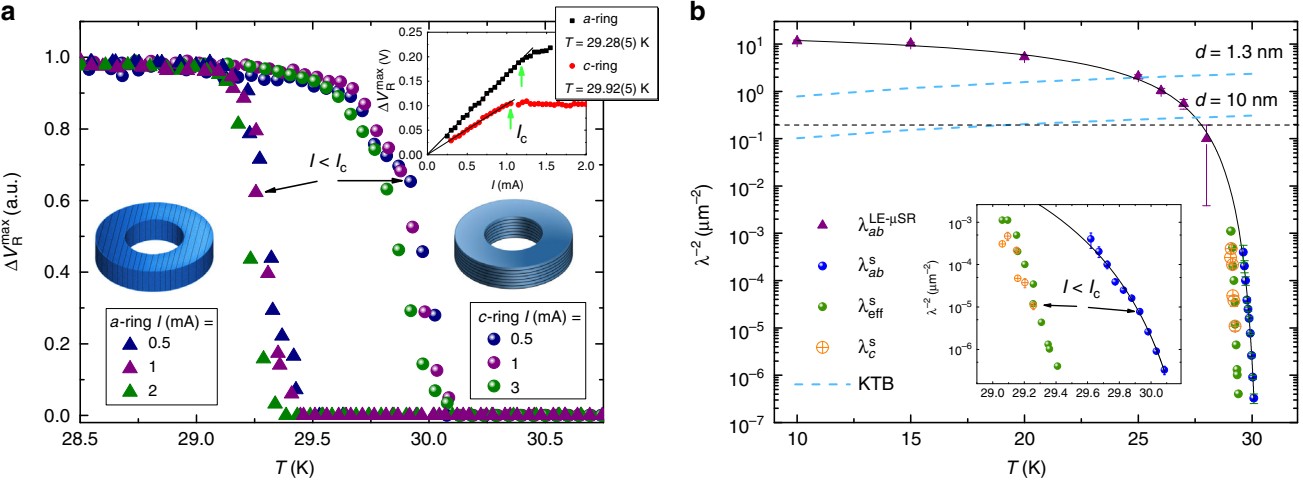

**Fig. 3** LSCO $x = 0.125$ stiffness. **a** Comparison between $a$- and $c$-rings, which are demonstrated in the figure, as measured by the Stiffnessometer. The signal is normalized by the maximum measured ring voltage. Different transition temperatures are observed for the two kind of rings with 0.7 K difference between them. The transition does not depend on the applied current in the inner coil up to 1 mA. The highest temperatures for which the current in the coil $I$ is clearly smaller than the critical current $I_c$ are marked. Inset: Critical current measurement $I_c$ of $c$-ring at $T = 29.92(5)$ K and $a$-ring at $T = 29.28(5)$ K. **b** Semi-log plot of $\lambda_{ab}^{-2}$ as measured by LE-$\mu$SR (purple solid triangles) and Stiffnessometer (blue solid spheres). Error bars are the results of the fitting procedures. Black dashed line represents the sensitivity limit of LE-$\mu$SR. Black solid line is a fit to a phenomenological function described in the text. Dashed cyan lines represent the KTB line for layer widths $d = 1.3$ nm and $d = 10$ nm. Green solid spheres represent the penetration depth of an $a$-ring from the Stiffnessometer, analyzed as if the ring is isotropic with $\lambda_{eff}$ which is some combination of $\lambda_{ab}$ and $\lambda_c$. Orange open symbols show $\lambda_c$ obtained at the temperature range where their ratio is manageable numerically for analysis. The inset is a zoom-in on temperatures close to the transitions. The highest temperatures for which clearly $I < I_c$ and the extraction of $\lambda$ is valid are marked

the sample surface, $x$, resulting from the Meissner effect. Thus, the muons spin frequency becomes smaller as they stop deeper in the sample. In this case, the asymmetry is given by

$$A(E, t) = A_0 e^{-t/u} \int_0^\infty p(x, E) \cos\left(\gamma B_0 e^{-x/\lambda} t\right) dx, \quad (5)$$

where $1/u$ represents contributions to the relaxation from depth independent processes (Supplementary Note 3) and $B_0$ is the magnetic induction outside of the sample and parallel to its surface. For our LE-$\mu$SR measurements, the sample is a mosaic of plates cut in the $ac$ crystallographic plane from the same LSCO $x = 0.125$ crystal used for the Stiffnessometer. Each plate was mechanically polished to a roughness of few tens of nanometers. The plates were glued to a nickel-coated plate using silver paint (see Fig. 4 inset). We cooled the sample to 5 K in zero magnetic field. Then a transverse magnetic field was applied along the $\hat{\mathbf{a}}$ or $\hat{\mathbf{c}}$ directions, and we warmed to the desired measurement temperature.

Figure 5 presents asymmetry data for both magnetic field orientations and different implantation energies. Panels (a) and (b) show data for $\mathbf{H} \parallel \hat{\mathbf{c}}$ at two different temperatures, and panel (c) depicts data for $\mathbf{H} \parallel \hat{\mathbf{a}}$. The data sets are shifted vertically for clarity. We limit the presentation to temperatures above 10 K, since below it strong relaxation due to spin density wave order obscures the oscillatory signal. At $T = 20$ K and $\mathbf{H} \parallel \hat{\mathbf{c}}$, we observe a clear frequency shift as a function of implantation energy, indicating a Meissner state. However, for $T = 30$ K, where the Stiffnessometer clearly shows $\rho_{ab} > 0$, we could not detect any change in frequency, even though we used high statistics data acquisition of 24 million events for $E = 23$ keV and 8 million for the rest. This can be explained by the fact that the penetration depth here is much longer than the muon stopping length scale of the order of hundred nanometers. When $\mathbf{H} \parallel \hat{\mathbf{a}}$ we did not observe any frequency shift at all temperatures, even though the sample is in the Meissner state.

We fit the LE-$\mu$SR asymmetry spectra to Eq. (5) and extract $\lambda_{ab}$, as shown in Fig. 3b. There is a gap between the available

data from the two techniques because the longest penetration depth that LE-$\mu$SR can measure, represented by the horizontal dashed line in the figure, is much smaller than the shortest $\lambda$ for which the Stiffnessometer is sensitive to. The function $\lambda_{ab}^{-2} = C_0 \exp\left[C_1 / \left(1 + C_3(1 - T/T_c)^\delta\right)\right]$ is fitted to the combined data and serves for interpolation. Since at $T = 10$ K we could only measure $\lambda_{ab}$ and not $\lambda_c$, we deduce an anisotropy $\lambda_c(0)/\lambda_{ab}(0) \geq 10$, as was observed in $\mu$SR, optical, and surface impedance measurements[15–17].

**Extracting $\lambda_c$.** We are now in position to extract $\lambda_c$ from Eqs. (2) and (3) and the Stiffnessometer $a$-ring data in Fig. 3a. In this case, two coupled partial differential equations must be solved, where $\lambda_{ab}$ is determined from the $c$-ring interpolation. Currently, we manage to extract $\lambda_c$ for only few temperatures close to $T_s^a$, where the anisotropy ratio is not too big and numerically solvable. These values of $\lambda_c$ are presented as orange open symbols in Fig. 3b. The extraction of $\lambda_c$ is done from our $I = 1$ mA measurement and is again valid only as long as $I < I_c$. This point is marked in the inset. As stated before, beyond that point the extraction only gives a qualitative trend of $\lambda_c(T)$. The SC currents in the ring at $z = 0$ emerging from the numerical solution for $T = 29.16$ K are depicted in Fig. 2c by combined contour and quiver plots.

For all $a$-ring Stiffnessometer data we also applied the $c$-ring stiffness extraction method ignoring the anisotropy. By doing so we determine an effective stiffness $\lambda_{eff}^{-2}$, which is some combination of $\lambda_{ab}^{-2}$ and $\lambda_c^{-2}$. These values are presented as green solid spheres in Fig. 3b. $\lambda_{eff}^{-2}$ is larger than $\lambda_c^{-2}$ but shows the same trend and indicates two transition temperatures.

## Discussion

The observation of two transition temperatures is awkward; a material should have only one SC critical temperature. One possible speculation for this result is a finite size effect, namely, if the rings could be made bigger the difference between the two

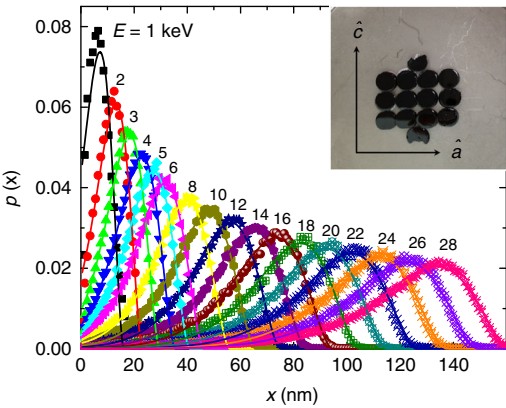

**Fig. 4** Muon stopping profiles. The probability distribution $p(x)$ of a muon to stop at some depth $x$ inside the sample for different implantation energies. The inset shows the LSCO $x = 0.125$ single crystal samples used in the experiment. All the pieces were polished to roughness of several nanometers. The crystallographic axes $a$ and $c$ are in the plane of the samples, and shown in the picture

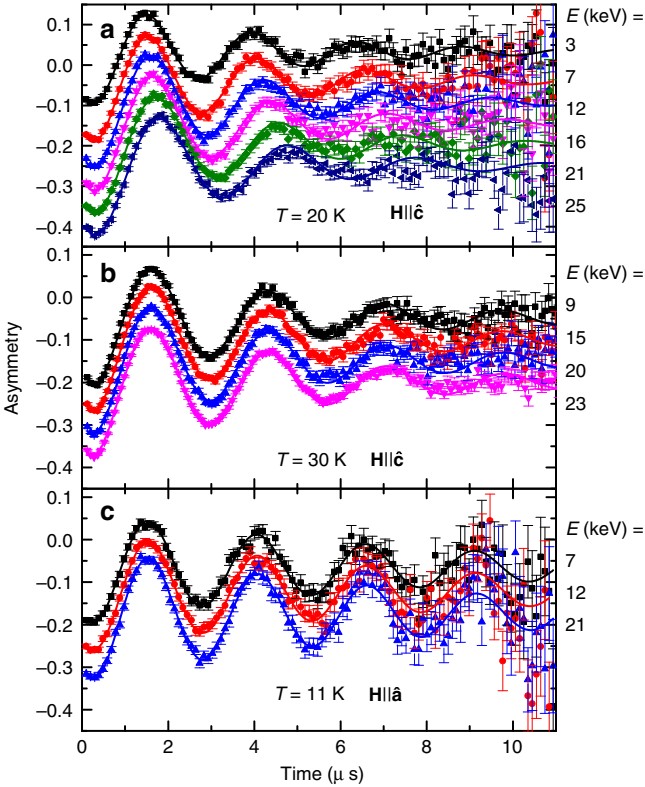

**Fig. 5** LE-$\mu$SR spectra. Asymmetry as a function of time for different muon implantation energies for: **a** $\mathbf{H} \parallel \hat{\mathbf{c}}$, $H = 26.7$ Oe, $T = 20$ K, **b** $\mathbf{H} \parallel \hat{\mathbf{c}}$, $H = 26.7$ Oe, $T = 30$ K, **c** $\mathbf{H} \parallel \hat{\mathbf{a}}$, $H = 26.3$ Oe, $T = 11$ K. A clear frequency shift as a function of implantation energy is observed in **a**. In the (**b**) conditions, the Stiffnessometer clearly detects stiffness in the $ab$ plane (Fig. 3a), while no frequency shift is observed by LE-$\mu$SR within our sensitivity. For $\mathbf{H} \parallel \hat{\mathbf{a}}$ (**c**) there is no frequency shift at all temperatures. Each data point in panels (**a**–**c**) represents high statistics data accumulation and the error bar is the standard deviation

transition temperatures would diminish. This, however, cannot be the case since the sample size is taken into account when extracting the stiffness. Bigger samples should lead to the same $\lambda$ values. A more plausible explanation is that the phase transition starts in the form of wide superconducting filaments[18] or finite width sheets[19] in the planes, but disconnected in the third direction. Whether this is the case, or our result indicates a new type of phase transitions, requires further and more local experiments.

The two transition temperatures suggest that there is a temperature range in which the system behaves purely as 2D. Therefore, we examine whether $\lambda_{ab}^{-2}$ follows the KTB behavior. At the KTB transition, the stiffness should undergo a sharp increase (a "jump") at a temperature $T_{KTB}$ that satisfies $\lambda^{-2} = \gamma T_{KTB}$, where $\gamma = \frac{8 k_B e^2 \mu_0}{\pi \hbar^2 d}$ and $d$ is the layer thickness[20]. We plot the line $\lambda^{-2} = \gamma T$ in Fig. 3b for thickness $d = 1.3$ nm of one unit cell (u.c.) and for $d = 10$ nm of about 8 u.c., both in cyan dashed lines. Clearly, the KTB line for thickness of one u.c. does not intersect $\lambda_{ab}^{-2}$ where it exhibits a jump. The line for $d = 10$ nm, however, does seem to intersect at the beginning of a jump. Thus, for the transition to be of the KTB nature, an effective layer of about 8 unit cells or more is needed.

In summary, using magnetic-field-free superconducting stiffness tensor measurements, which are sensitive to long penetration depths, on the order of millimeters, and which are not affected by demagnetization factors or vortices, we shed light on the SC phase transition in LSCO $x = 1/8$. In this compound, there is a temperature interval of 0.7 K where SC current can flow in the $CuO_2$ planes but not between them.

## Methods

**Materials**. The LSCO single crystals were grown using Traveling Solvent Floating Zone furnace, annealed in Argon environment at $T = 850$ C for 120 h to release internal stress, and oriented by Laue X-ray diffraction. Stiffnessometer samples were cut into a shape of rings using pulsed Laser ablation, after which the rings were annealed again. LE-$\mu$SR samples were mechanically polished using diamond paste. They were treated eventually with 20 nm alumina suspension. The resulting roughness of few tens of nanometers was determined by atomic force microscope (AFM). A typical AFM data are presented in Supplementary Fig. 1.

**Stiffnessometer**. The Stiffnessometer is an add-on to a cryogenic SQUID magnetometer. The components of the experiment shown in Fig. 2 in the main text are as follows: The inner coil is 60 mm long with a 0.05 mm diameter wire and two layers of windings. It is wound on top of a 0.54 mm diameter polyamide tube. The outer diameter of the coil is 0.74 mm, and it has 40 turns per millimeter. The

second-order gradiometer is 14 mm high, with inner diameter 25.9 mm, outer diameter 26.3 mm, and made from 0.2 mm diameter wire. We take $R_{PL} = 13 \pm 0.15$ mm. The gradiometer is constructed from three groups of windings distanced 7 mm apart from each other. The upper and lower ones have two loops wound clockwise, while the center windings have four loops wound anticlockwise. Numeric evaluation of the $G$ factor in Eq. (2) using the gradiometer dimensions gives a reasonable result for an isotropic superconducting ring with known dimensions[12].

For anisotropic ring the situation is much more complicated. Therefore, the $G$ factor used here is extracted experimentally. As shown in Fig. 3, the signal from the rings $\Delta V_R^{max}$ saturates at $T \ll T_c$. It happens when the penetration depth is much smaller than the ring dimensions. The ratio between the voltages saturation value to the vector potentials ratio calculated numerically gives $G$.

**LE-$\mu$SR**. In the LE-$\mu$SR experiment, 4 MeV spin-polarized muons are stopped at a moderator, made of 300-nm-thick layer of solid Argon grown on top of a silver foil. They are then accelerated to a chosen energy between 1 and 30 keV by applying a voltage difference between the foil and the sample. The sample holder is placed on a sapphire plate hence electrically isolated. The parameters of the stopping profiles given in the main text by Eq. (4) are

$$p_0(E) = \exp\left[-6.4 - 0.8 \ln E - 0.18 (\ln E)^2\right]$$
$$x_0(E) = 12 + 6E - 0.11 E^2 + 0.0028 E^3$$
$$\zeta(E) = 2.77 + 0.49 E - 0.0165 E^2 + 0.0003 E^3.$$

The whole beam line is under ultra high vacuum of $\sim 10^{-10}$ mbar, and the stopping and accelerating processes of the muons preserve most of the polarization. Once in the sample, the muon spin precesses in the local external or internal magnetic field and the time-dependent polarization is reconstructed from asymmetry in the positrons decay, which are emitted preferentially in the muon spin direction at the time of decay.

## Data availability

The raw LE-$\mu$SR spectra are available at http://musruser.psi.ch/cgi-bin/SearchDB.cgi, under "LSCO, $x = 0.125$". The rest of the data that support the findings of this study are available from the corresponding author upon reasonable request.

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

## Acknowledgements

The Technion physics team is supported by the Israeli Science Foundation (ISF) and by the Technion RBNI Nevet program. The LE-$\mu$SR work is based on experiments performed on the LEM beam line[21] at the Swiss Muon Source S$\mu$S, Paul Scherrer Institute, Villigen, Switzerland. We are grateful for helpful discussions with Boris Shapiro, Assa Auerbach, Daniel Podolsky, Andreas Suter, and Ori Scally.

## Author contributions

A.K. conceived the Stiffnessometer and supervised the project, A.K. and I.K. developed the Stiffnessometr technique, I.K. and I.M. performed the Stiffnessometer measurements, I.K. grew and characterized the samples, I.K., Z.S., A.K. and T.P. performed the LE-$\mu$SR measurements and data analysis, I.K., N.G. and A.K. performed the Stiffnessometer data analysis, I.K. and A.K. wrote the manuscript in consultation with all authors.

## Additional information

**Competing interests:** The authors declare no competing interests.

