## [Peer Review File · Nature Communications]

Reviewers' comments:

Reviewer #1 (Remarks to the Author):

This is an interesting and original work as far as experiment is concerned. The idea to subject a superconducting ring to an applied non-zero vector potential A while keeping the field $H=0$ is intriguing.

However, just because it is original and novel, unfortunately the authors left a number of points unclear, that makes the work less convincing. Some of those, questionable in my view or not explained sufficiently, are listed below:

1. It is not clear whether or not the field H is screened to $H=0$ outside the inner coil in the location of the ring-sample. The non-zero response in the normal state at $T>T_c$, shown in the inset to Fig. 2b, seem to suggest that this is not the case.

2. It is not clear why the authors claim (see their Ref.12 on the same subject) that their "method determines ρ_s directly without the use of the penetration depth concept". In Eq.(3) of the same text the stiffness is defined as $\rho_s = c/4\pi \lambda^2$. Hence, the 'stiffness' is proportional to λ^{-2} with the universal coefficient, i.e. physically they contain the same information. It is just more convenient to plot $1/\lambda^2$ than a divergent at T_c λ^2 , hence the concept of stiffness is popular.

3. Still, as it is claimed in Appendix on Numerical methods, the condition $H_{\text{applied}} = 0$ at the ring surface is employed in formulating the boundary conditions on the vector potential. My difficulty is that even if $H_{\text{applied}} = 0$, the ring currents create non-zero magnetic field outside and inside the ring. Then the main claim in the title of having "magnetic field free stiffness meter" sounds confusing.

4. I don't quite understand why one can disregard the phase ϕ in the London eq-n $J \sim \rho_s (\text{grad}\phi - A)$ in a superconducting ring. In Ref. 12, the authors say that their sequence of cooling in zero field, ZGFC of preparing the state they study, sets $\text{grad}\phi=0$, because the integer of vorticity should be preserved. I do not see the reason for such conservation.

5. The μSR part of the paper describes an interesting method to extract the penetration depth $\lambda(T)$. The μSR data are compatible with those obtained with author's Stiffnessometer at temperatures close to T_c with $\lambda \sim \text{mm}$, i.e., $\lambda^{-2} \sim 10^{-6} \text{ micron}^{-2}$ shown in Fig.3b. My problem is that within mean-field theory (such as GL or London) $[\lambda(0)/\lambda]^2$ goes as $(T_c-T)/T_c$. Taking $\lambda(0)^{-2} \sim 10 \text{ micron}^{-2}$ from the fig, we get $(T_c-T)/T_c \sim 10^{-7}$. This is exceedingly close to T_c and for a cuprate is likely to be in the domain of critical fluctuations, where mean-field based models should not be used.

In my view, despite the serious effort the authors made in collecting the data, the interpretation is questionable. Hence, the paper in the present form is not convincing and can cause a lot of confusion. I do not recommend it for publication.

Reviewer #2 (Remarks to the Author):

Kapon et al introduce a method to measure superconducting stiffness. Further to in-house experiments, the authors perform complementary low energy μSR experiments to determine the magnetic penetration depth and therefore discuss trends in the superfluid density.

First, this is a clever in-house experimental set up addressing a fundamental physical parameter. It is a simple, convenient and potentially useful approach. I anticipate considerable interest in its

utilization and of course further development by the scientific community.

Second, the authors demonstrate an understanding of the basic physical and technical principles (including limitations) and deliver a systematic investigation, starting from elaborate sample preparation, characterisation and data analysis.

Third, the work uses a prototypical and well characterised high-T_c cuprate to address a topic of central interest in the general subject of superconductivity, with focus on low dimensional superconductors. La-Sr-Cu-O has been subject of investigation for more than 3 decades, giving insight to key questions in the study of unconventional superconductivity. Kapon et al go beyond incremental contribution. They introduce a method to discern fundamental differences in the behaviour of a basic physical quantity namely, superconducting stiffness.

Fourth, the authors appreciate the assumptions in the analysis and subsequent ambiguities in the trends, and eventual conclusions of the work. However, crucially Figure 3 depicts effects, which add credence to relevant literature indicating validity of the methodology. Importantly, the authors can identify distinct signatures on the role of dimensionality in the onset of a superconducting phase transition in layered cuprates. In particular, a temperature range with 2D like stiffness. This important observation is subject of increasing investigations by the community.

In summary, the work of Kapon et al is a beautiful contribution in the study of superconductivity. Their observation of two transition temperatures and a finite range in temperature where superconducting currents flow only in one crystallographic direction indicates tendency for decoupling of layers and potential for distinct superconducting properties in bulk single crystals. Wide and early dissemination of this work will motivate further studies and improvements, both in the technique and the analysis for higher resolution results and the application of the reported methodology to many other superconducting systems for similar investigations such as the physics of anisotropic superconductors in a region of electronic heterogeneity, including stripes.

I recommend publication in Nature Communications.

Reviewer #3 (Remarks to the Author):

This manuscript describes measurements with a novel device sensitive to small magnetizations. This so-called "Stiffnessmeter" uses a SQUID magnetometer to detect a small super current in a toroid of LSCO in the temperature range near T_c. The authors also use low energy muon spin rotation to measure the superfluid density at temperatures much lower than T_c. The main conclusion of this manuscript is that there is a difference of about 0.7K in the superconducting T_c's as determined from the superfluid density (stiffness) between samples where the super currents flow in the ac plane and the plane normal to the c-axis. The authors interpret this difference to the 2-dimensional nature of the underlying crystal structure and electronic states.

My main criticisms of this manuscript are that the paper spends a lot of time discussing the details of their new device but do not really address some of the experimental details. For example, since the measurements of stiffness are done on two different specimens due to the requirement to drill a hole in them in different directions, to what extent are the samples identical? For example, do standard squid measurements of T_c, with the field applied in the same crystalline direction give the same T_c? The theory describing the operation of the device refers to an infinitely long solenoid such that the field outside it is zero. Given the finite (though long) length of the solenoid, to what extent does this matter?

Reply to referees

In what follows, we answer the points one by one, and provide the missing information and explain where in the manuscript corrections have been made:

Reviewer #1

This is an interesting and original work as far as experiment is concerned. The idea to subject a superconducting ring to an applied non-zero vector potential A while keeping the field $H=0$ is intriguing.

However, just because it is original and novel, unfortunately the authors left a number of points unclear, that makes the work less convincing. Some of those, questionable in my view or not explained sufficiently, are listed below:

1. It is not clear whether or not the field H is screened to $H=0$ outside the inner coil in the location of the ring-sample. The non-zero response in the normal state at $T>T_c$, shown in the inset to Fig. 2b, seem to suggest that this is not the case.

The “non-zero response in the normal state at $T>T_c$, shown in the inset to Fig. 2b” is the response only from the edges of the coil. These can be thought of as point-like Dirac monopoles, since magnetic field does leak from the edges. However, as explained in the text of Ref. 12, this leakage is zero or almost zero in the ring area. Any remnant magnetic field at the ring area, usually on the order of few milligauss, is canceled with an external coil. To prove this, we performed tests with an open ring. The signal from an open ring is due to magnetic field and the vector potential since the current cannot close a loop. The high quality of the results of this procedure are demonstrated in Fig. 4 in Ref. 12. In the new version we clarify this point by adding the sentence: “The field at the ring position is measured with an open ring where supercurrent cannot close a loop, hence any signal is due to the field only. This procedure is described in detail in Ref. [12]”

2. It is not clear why the authors claim (see their Ref.12 on the same subject) that their “method determines ρ_s directly without the use of the penetration depth concept”. In Eq.(3) of the same text the stiffness is defined as $\rho_s = c/4\pi \lambda^2$. Hence, the ‘stiffness’ is proportional to λ^{-2} with the universal coefficient, i.e. physically they contain the same information. It is just more convenient to plot $1/\lambda^2$ than a divergent at $T_c \lambda^2$, hence the concept of stiffness is popular.

We claim that the method determines ρ_s without the use of the penetration depth concept because the sample does not experience an externally applied magnetic field which decays into it. It is through that if one would measure the penetration depth with an externally applied field, one would find that his λ and our ρ_s agree with each other

according to the equation given above. But, since we do not apply a field, we do not measure its penetration.

3. Still, as it is claimed in Appendix on Numerical methods, the condition $H_{\text{applied}} = 0$ at the ring surface is employed in formulating the boundary conditions on the vector potential. My difficulty is that even if $H_{\text{applied}} = 0$, the ring currents create non-zero magnetic field outside and inside the ring. Then the main claim in the title of having “magnetic field free stiffness meter” sounds confusing.

The referee is correct, and the ring does create its own field. But, as in the book of Jackson, e.g., we distinguish between two fields: H - the applied magnetic field, and B - the magnetic induction. The Stiffnessometer works in zero applied magnetic field, $H=0$, which is the field set by external currents. In addition, in the weak stiffness limit $H=B=0$ and the self-field of the ring on itself is a second order effect. Therefore, close to T_c both the magnetic field and the magnetic induction can be approximated by zero. This is demonstrated in Ref. 12 inset of Fig. 8. To make this point clearer we changed the abstract to read “external magnetic field” rather than “magnetic field”.

4. I don't quite understand why one can disregard the phase ϕ in the London equation $J \sim \rho_s (\text{grad}\phi - A)$ in a superconducting ring. In Ref. 12, the authors say that their sequence of cooling in zero field, ZGFC of preparing the state they study, sets $\text{grad}\phi=0$, because the integer of vorticity should be preserved. I do not see the reason for such conservation.

The phase ϕ can be viewed as an in-plane arrow. Cooling at $A=0$ must set $\nabla\phi=0$ to minimize the energy, which is proportional to the current squared. Namely, all arrows point in the same direction. Since the phase is quantized, to change $\nabla\phi$ means to make a twist of all arrows in a closed loop, such that the phase between the first and last arrows change by 2π . This would lead to a discontinuity in the value of the phase, a procedure that costs energy. A nice analog is a ferromagnetic chain with the spins pointing in the same direction. To rotate the first spin with respect to the last spin by 2π requires breaking a bond. Thus, it is energetically not favorable for a ferromagnet or the superconducting ring. Therefore, when turning A on after cooling, all “arrows” continue to point in the same direction and $\nabla\phi=0$, until A exceeds a critical value. At this point the current is too high and it is worth while for the superconductor to “break a bond” and reduce the current.

An experimental proof for that is given in Ref. 12 inset of Fig. 5. The measured voltage, which is proportional to the current J , is proportional to the coil current, which, in turn, is proportional to A . This proportionality sequence means that $\nabla\phi=0$ for all coil currents up to a critical value.

Since this discussion is on Ref. 12 we made no changes to the present manuscript.

5. The muSR part of the paper describes an interesting method to extract the penetration depth $\lambda(T)$. The muSR data are compatible with those obtained with author's Stiffnessometer at temperatures close to T_c with $\lambda \sim \text{mm}$, i.e., $\lambda^{-2} \sim 10^{(-6)}$ μm^{-2} shown in Fig.3b. My problem is that within mean-field theory (such as GL or London) $[\lambda(0)/\lambda]^2$ goes as $(T_c-T)/T_c$. Taking $\lambda(0)^{-2} \sim 10$ μm^{-2} from the fig, we get $(T_c-T)/T_c \sim 10^{(-7)}$. This is exceedingly close to T_c and for a cuprate is likely to be in the domain of critical fluctuations, where mean-field based models should not be used.

It is hard for us to understand this comment. Our work is experimental and we provide measured values of λ^{-2} . We don't claim that $\lambda^2(0)/\lambda^2(T)$ goes as $(T_c-T)/T_c$. In fact, it does not. We also did not use a mean-field-based models or any other model. To extract data from the stiffnessometer, we only used Maxwell's and London's equations, which we are sure the referee is not contesting.

Reviewer #3

This manuscript describes measurements with a novel device sensitive to small magnetizations. This so-called "Stiffnessmeter" uses a SQUID magnetometer to detect a small super current in a toroid of LSCO in the temperature range near T_c . The authors also use low energy muon spin rotation to measure the superfluid density at temperatures much lower than T_c . The main conclusion of this manuscript is that there is a difference of about 0.7K in the superconducting T_c 's as determined from the superfluid density (stiffness) between samples where the super currents flow in the ac plane and the plane normal to the c-axis. The authors interpret this difference to the 2-dimensional nature of the underlying crystal structure and electronic states.

My main criticisms of this manuscript are:

1. that the paper spends a lot of time discussing the details of their new device but do not really address some of the experimental details. For example, since the measurements of stiffness are done on two different specimens due to the requirement to drill a hole in them in different directions, to what extent are the samples identical? For example, do standard squid measurements of T_c , with the field applied in the same crystalline direction give the same T_c ?

We indeed performed magnetization measurements on these crystals. An example is presented in the inset of Fig. 1. More measurements are presented in Ref. [8]. The measurements were done on needle shaped samples cut from the same LSCO rod. The needle shape is used to eliminate demagnetization factor effects. They are different in the path the current must take to expel the magnetic field. In c-needles the current run only in the CuO_2 planes, in the a-needles the current must cross planes. We found systematic difference in T_c for several samples. However, as we explain in the introduction,

magnetization is different from stiffness and is less demanding. Hence the stiffnessometer experiment is more informative but agrees with the needles measurements.

2. The theory describing the operation of the device refers to an infinitely long solenoid such that the field outside it is zero. Given the finite (though long) length of the solenoid, to what extent does this matter?

We addressed this question in our reply to reviewer 1. Here we emphasize our point again. We performed numerous tests to assure that there is NO magnetic field anywhere in the ring area. This is done using an open ring, that is sensitive only to an applied magnetic field but not to an applied vector potential. Any remanent magnetic field, usually on the order of few milligauss, is canceled with the external coil. The high quality zero field obtained with this procedure is demonstrated in Fig. 4 of Ref. 12.

Second, we calculated the vector potential of finite coil (Fig. 6 of Ref. 12). It is clear that A of our finite coil falls off like $1/r$ outside the coil and in the ring area, exactly like infinite coil does. This is the only important property of the coil, since the absolute magnitude of the applied A is measured by ΔV_{IC}^{\max} .

Reply to referees

We would like to thank all three referees for taking the time to read our manuscript, and Ref. 12, which contains technical details. We have been developing our technique for several years and have become oblivious to some details. In our response to the referees, we provide the missing information and explain where in the manuscript corrections have been made.

Reviewer #1

This is an interesting and original work as far as experiment is concerned. The idea to subject a superconducting ring to an applied non-zero vector potential A while keeping the field $H=0$ is intriguing.

However, just because it is original and novel, unfortunately the authors left a number of points unclear, that makes the work less convincing. Some of those, questionable in my view or not explained sufficiently, are listed below:

1. It is not clear whether or not the field H is screened to $H=0$ outside the inner coil in the location of the ring-sample. The non-zero response in the normal state at $T>T_c$, shown in the inset to Fig. 2b, seem to suggest that this is not the case.

The “non-zero response in the normal state at $T>T_c$, shown in the inset to Fig. 2b” is the response from the edges of the coil only. These can be thought of as point-like Dirac monopoles, since magnetic field does leak from the edges. However, as explained in the text of Ref. 12, this leakage is zero or almost zero in the ring area. Any reminiscent magnetic field at the ring area, usually on the order of few milligauss, is canceled with an external coil. To prove this, we performed tests with an open ring. The signal from an open ring is due to magnetic field and not the vector potential since the current cannot close a loop. The high quality of the results of this procedure are demonstrated in Fig. 4 in Ref. 12. In the new version we clarify this point by adding to the main text the sentence: “The field at the ring position is measured with an open ring where supercurrent cannot close a loop, hence any signal is due to the field only. This procedure is described in detail in Ref. [12]...”. We also added sections B.1. and B.2. to the Supplementary Materials, discussing the effect of finite coil and the extent to which the magnetic field outside of it is zero.

2. It is not clear why the authors claim (see their Ref.12 on the same subject) that their “method determines ρ_s directly without the use of the penetration depth concept”. In Eq.(3) of the same text the stiffness is defined as

$\rho_s = c/4\pi \lambda^2$. Hence, the ‘stiffness’ is proportional to λ^{-2} with the universal coefficient, i.e. physically they contain the same information. It is just more convenient to plot $1/\lambda^2$ than a divergent at $T_c \lambda^2$, hence the concept of stiffness is popular.

We claim that the method determines ρ_s without the use of the penetration depth concept because the sample does not experience an externally applied magnetic field which decays into it. It is through that if one would measure the penetration depth with an externally applied field, one would find that his λ and our ρ_s agree with each other according to the equation given above. But, since we do not apply a field, we do not measure its penetration.

3. Still, as it is claimed in Appendix on Numerical methods, the condition $H_{\text{applied}} = 0$ at the ring surface is employed in formulating the boundary conditions on the vector potential. My difficulty is that even if $H_{\text{applied}} = 0$, the ring currents create non-zero magnetic field outside and inside the ring. Then the main claim in the title of having “magnetic field free stiffness meter” sounds confusing.

The referee is correct, and the ring does create its own field. But, as in the book of Jackson, e.g., we distinguish between two fields: H - the applied magnetic field, and B - the magnetic induction. The Stiffnessometer works in zero applied magnetic field, $H=0$, which is the field set by external currents. In addition, in the weak stiffness limit $H=B=0$ and the self-field of the ring on itself is a second order effect. Therefore, close to T_c both the magnetic field and the magnetic induction can be approximated by zero. This is demonstrated in Ref. 12 inset of Fig. 8. To make this point clearer we changed the abstract to read “external magnetic field” rather than “magnetic field”.

4. I don’t quite understand why one can disregard the phase ϕ in the London eq-n $J \sim \rho_s (\text{grad}\phi - A)$ in a superconducting ring. In Ref. 12, the authors say that their sequence of cooling in zero field, ZGFC of preparing the state they study, sets $\text{grad}\phi=0$, because the integer of vorticity should be preserved. I do not see the reason for such conservation.

The phase ϕ can be viewed as an in-plane arrow. Cooling at $A=0$ must set $\nabla\phi=0$ to minimize the energy, which is proportional to the current squared. Namely, all arrows point in the same direction. Since the phase is quantized, to change $\nabla\phi$ means to make a twist of all arrows in a closed loop, such that the phase between the first and last arrows changes by 2π . This would lead to a discontinuity in the value of the phase, a procedure that costs energy. A nice analog is a ferromagnetic chain with the spins pointing in the same direction. To rotate the first spin with respect to the last spin by 2π requires breaking a bond. Thus, it is energetically not favorable for a ferromagnet (or the superconducting ring). Therefore, when turning A on after cooling, all “arrows” continue to point in the same direction and $\nabla\phi=0$, until A exceeds a critical value. At this point the

current in the SC is too high and it is worth while for the superconductor to “break a bond” and reduce the current.

An experimental proof for that is given in Ref. 12 inset of Fig. 5. The measured voltage, which is proportional to the current J , is proportional the coil current, which, in turn, is proportional to A . This proportionality sequence means that $\nabla\varphi = 0$ for all coil currents up to a critical value.

To make this point clearer in our paper, we added section B.3. to the Supplementary Materials, discussing the validity of London’s equation to our experiment.

5. The muSR part of the paper describes an interesting method to extract the penetration depth $\lambda(T)$. The muSR data are compatible with those obtained with author’s Stiffnessometer at temperatures close to T_c with $\lambda \sim \text{mm}$, i.e., $\lambda^{-2} \sim 10^{(-6)}$ micron⁽⁻²⁾ shown in Fig.3b. My problem is that within mean-field theory (such as GL or London) $[\lambda(0)/\lambda]^2$ goes as $(T_c-T)/T_c$. Taking $\lambda(0)^{-2} \sim 10$ micron⁽⁻²⁾ from the fig, we get $(T_c-T)/T_c \sim 10^{(-7)}$. This is exceedingly close to T_c and for a cuprate is likely to be in the domain of critical fluctuations, where mean-field based models should not be used.

It is hard for us to understand this comment. Our work is experimental and we provide measured values of λ^{-2} . We don’t claim that $\lambda^2(0)/\lambda^2(T)$ goes as $(T_c-T)/T_c$. In fact, it does not. We also did not use a mean-field-based models or any other model. To extract data from the stiffnessometer, we only used Maxwell’s and London’s equations, which we are sure the referee is not contesting.

Reviewer #3

This manuscript describes measurements with a novel device sensitive to small magnetizations. This so-called "Stiffnessmeter" uses a SQUID magnetometer to detect a small super current in a toroid of LSCO in the temperature range near T_c . The authors also use low energy muon spin rotation to measure the superfluid density at temperatures much lower than T_c . The main conclusion of this manuscript is that there is a difference of about 0.7K in the superconducting T_c 's as determined from the superfluid density (stiffness) between samples where the super currents flow in the ac plane and the plane normal to the c-axis. The authors interpret this difference to the 2-dimensional nature of the underlying crystal structure and electronic states.

My main criticisms of this manuscript are:

1. that the paper spends a lot of time discussing the details of their new device but do not really address some of the experimental details. For example, since the measurements of stiffness are done on two different specimens due to the requirement to drill a hole in

them in different directions, to what extent are the samples identical? For example, do standard squid measurements of T_c , with the field applied in the same crystalline direction give the same T_c ?

We indeed performed magnetization measurements on these crystals. An example is presented in the inset of Fig. 1. More measurements are presented in Ref. [8]. The measurements were done on needle shaped samples cut from the same LSCO rod. The needle shape is used to eliminate demagnetization factor effects. They are different in the path the current must take to expel the magnetic field. In c-neededles the current run only in the CuO_2 planes, in the a-neededles the current must cross planes. We found systematic difference in T_c for several samples. However, as we explain in the introduction, magnetization is different from stiffness and is less demanding. Hence the stiffnessometer experiment is more informative but agrees with the needles measurements.

2. The theory describing the operation of the device refers to an infinitely long solenoid such that the field outside it is zero. Given the finite (though long) length of the solenoid, to what extent does this matter?

We addressed this question in our reply to reviewer 1. Here we emphasize our point again. We performed numerous tests to assure that there is NO magnetic field anywhere in the ring area. This is done using an open ring, that is sensitive only to an applied magnetic field but not to an applied vector potential. Any remniscent magnetic field, usually on the order of few milligauss, is canceled with the external coil. The high quality zero field obtained with this procedure is demonstrated in Fig. 4 of Ref. 12 and supplementary B.1.

Second, we calculated the vector potential of finite coil (Fig. 6 of Ref. 12 and now supplementary B.2.). It is clear that A of our finite coil falls off like $1/r$ outside the coil and in the ring area, exactly like infinite coil does. This is the only important property of the coil, since the absolute magnitude of the applied A is measured by ΔV_{IC}^{\max} .

In the new version we clarify this point by adding to the main text the sentence: "The field at the ring position is measured with an open ring where supercurrent cannot close a loop, hence any signal is due to the field only. This procedure together with discussion on the coil being finite are described in detail in Ref. [12] and reviewed in Supplementary B.1. and B.2."

Reviewers' comments:

Reviewer #4 (Remarks to the Author):

This work reports measurements of the superfluid density $n_s(T)$ and the phase stiffness in LSCO rings near T_c using a novel "stiffnessometer" technique developed by the authors. The idea of subjecting a superconducting ring to an applied vector potential $\{\mathbf{A}\}$ at zero external magnetic field to measure small values of $n_s(T)$ near T_c is quite nice, although it may not be the only technique to do so. For instance, if a ring or a film stripline is a part of a resonant circuit, $n_s(T)$ can be inferred from the shift of the resonance frequency affected by the divergent kinetic inductance $\lambda^2(T)/d$ at $T \rightarrow T_c$.

I share the concern of other referees about the validity of the assumption made by the authors that the winding number of the order parameter l in the ring remains zero after the magnetic flux is changed. Because of fluxoid quantization, the state with zero phase gradient ($l=0$) has the lowest energy only if the ring is threaded by the flux Φ smaller than the flux quantum ϕ_0 . The latter is not the case for this experiment in which the double-layer winding with 40 turns/mm carrying 1 mA produces the field ≈ 1 Oe and $\Phi \approx 10^4 \phi_0$. In this macroscopic limit the phase gradient $\nabla \theta$ must be taken into account as it nearly compensates A in the bulk of the ring, so that the superfluid velocity $\{\mathbf{v}\}_s = (\hbar/m)(\nabla \theta - 2\pi i \{\mathbf{A}\}/\phi_0)$ of circulating currents vanishes if $\Phi = l\phi_0$ with $l \gg 1$. Because the ring is thicker than the penetration depth λ , minimization of the energies of supercurrents and self-field would result in supercurrents flowing in a thin layer at the surface.

I am not sure that the London model is adequate at temperatures close to T_c where the Ginzburg-Landau equations would be more appropriate. For instance, at $l=0$ the superfluid velocity $2\pi \hbar A/m \phi_0$ exceeds the pairbreaking limit $\hbar/m \xi(T)$ if $H > H_c \approx \phi_0/n^2 R \xi \approx (1 - T/T_c)^{1/2}$ [Oe]. This condition is satisfied for the parameters estimated above and $T/T_c > 0.98$ in the crucial region where the stiffnessometer signal shown in Fig 3 becomes strongly temperature-dependent. Here the analysis of experimental data can be complicated by penetration of vortices giving rise to big uncertainties in the penetration depth tensor. Concentration of the Meisner current density $J(r)$ at the surface shown in Fig. 12b facilitates penetration of vortices, although I could not use these data to estimate $J(r)$ as the factor $c/4\pi$ in CGS or μ_0 in SI is missing in the normalization on the vertical axis in Fig. 12b. Furthermore, preferential penetration of Josephson vortices along the ab planes in the layered LSCO might be one of the reasons of vanishing interlayer phase stiffness below T_c shown in Fig. 3. The observation that the interlayer phase coherence breaks down at $T < T_c$ is interesting, but in my opinion the London model used for the interpretation of this experiment may not be a good quantitative tool for the extraction of the superconducting phase stiffness.

Reply to referee 4:

We thank Ref. 4 for reading our manuscript extremely carefully and even performing calculations to test our results.

The referee accepts our main finding and say “The observation that the interlayer phase coherence breaks down at $T < T_c$ is interesting”. He even comes with an explanation to our main conclusion: “Josephson vortices along the ab planes in the layered LSCO might be one of the reasons of vanishing interlayer phase stiffness below T_c shown in Fig. 3”. Yet, surprisingly, he rejects an experimental paper on the basis of the analysis, which is not relevant to the main observation. We believe that even if the analysis can be done better, there is still validity to our experimental findings. However, as we argue below, our analysis is also valid considering the available data.

If, as the referee claims, “phase gradient $\nabla \theta$ must be taken into account as it nearly compensates A in the bulk of the ring” than the equation $\mathbf{v}_s = (\hbar/m)(\nabla \theta - 2\pi \mathbf{A}/\phi_0)$, should lead to \mathbf{v}_s as small as possible in the ring and we should have no signal from the ring at all, since there would be nearly no current in the ring. We are not sensitive to a current producing a single or even few flux quanta. Moreover, the signal should not be proportional to the applied current in the coil, and there should be no critical applied current. Our current dependent experiments presented in the Supplementary Fig. 10 (and in Ref. 12 Fig. 5) show that our signal is proportional to the applied current up to a critical value. For this reason, we neglect $\nabla \theta$ in our analysis up to the critical applied current.

The phase θ can be viewed as an in-plane arrow. Cooling at $A=0$ must set $\nabla \theta = 0$ to minimize the energy. Namely, all arrows point in the same direction. Since the phase is quantized, to change $\nabla \theta$ means to make a twist of all arrows in a closed loop, such that the phase between the first and last arrows on the loop changes by 2π . This would lead to a discontinuity in the value of the phase, a procedure that costs energy. A nice analog is a ferromagnetic ring with the spins pointing in the same direction. To rotate the last spin with respect to the first spin by 2π requires breaking a bond. It is energetically not favorable for a ferromagnet (or the superconducting ring) to break a bond. Therefore, when turning A on after cooling, all “arrows” continue to point in the same direction and $\nabla \theta = 0$, until A exceeds a critical value. At this point the current in the SC is too high and it is worthwhile for the superconductor to “break a bond” and reduce the current.

However, had we first applied the current in the coil and only than cooled the sample, a procedure we call Gauge Field Cooling (GFC), then the superconductor would have had a chance to choose $\nabla \theta$ such that $\mathbf{v}_s = 0$. In the GFC procedure, the referee is correct. Indeed, when we do the experiment in the GFC procedure we get no signal.

As for the London theory, it is valid when ξ is much smaller than λ . We measured ξ as a function of temperature in Ref. 12 Fig. 9 using the relations suggested by the referee. We found ξ to be smaller than λ by orders of magnitude at all temperatures. In this case, there is no need to invoke the full Ginzburg-Landau equations to extract λ from the measurements. Put in other words, our determination of λ was done with applied currents much smaller than the critical current at all temperatures, or, in the referee’s terminology, with superfluid velocity smaller than the pair braking one. So, our determination of λ is valid at all our temperatures.

These explanations have been added to the new manuscript by two new sentences in the abstract, two new sentences in the third paragraph on page 2, and an additional long paragraph in Supplementary B.3.

We would also like to point out that the referee suggests that there are other ways to measure λ . This is indeed correct. But we have the only way to measure λ without applying a magnetic field. In resonance circuits the sample causes frequency shift because it experiences a magnetic field. In this respect our method is unique.

We will appreciate if the referee can approve our manuscript as is mainly because he/she accept the main premise of the work and allow more profound analysis to be done in the future if the data will call for it.

Sincerely,

Itzik Kapon and Amit Keren

Reviewers' comments:

Reviewer #4 (Remarks to the Author):

As I stated in my previous report, I have no objections to this interesting experiment. My concern was and still is that the authors have made an important physical conclusion about vanishing phase stiffness in LSCO below T_c by using the London model in the analysis of their data. However, this conclusion is on shaky ground because the London model becomes inadequate at $T \rightarrow T_c$, no matter how small the field/current in the driving coil is. At $T = T_c$ the critical superfluid velocity vanishes, the induced current densities $J(x,y,z)$ in the ring are not small as compared to the depairing current density $J_d(T)$ so currents produce Josephson vortices penetrating along the ab planes. Generation of vortices at zero applied field is an interesting effect by itself, but it is very different from the scenario of the authors that the superfluid density vanishes below T_c at zero current. So the range of temperatures where the London approximation breaks down should be evaluated

1. I accept the argument of the authors that under the conditions of this experiment the phase of the order parameter can be set to zero.
2. In addition to $\lambda \gg \xi$, the applicability of the London model also requires the superfluid velocities v to be much smaller than the critical velocity $v_c = \Delta/p_F$, or the components $J_x(x,y)$ and $J_y(x,y)$ to be much smaller than J_d along the respective crystal axis. For an isotropic case in the gauge with zero phase, this implies small vector potentials, $A \ll A_c = \phi_0/2\sqrt{3}\pi\xi(T)$ (see Tinkham's book). I could not estimate the actual values of J_x and J_y from the plots given by the authors as Fig. 2c shows current streamlines in false color and unspecified units, while Fig 12 misses the speed of light or μ_0 in the units of current (see my first report). The numerical results shown in Fig. 11 suggest the maximum magnitude $A_m \approx 20\Phi/2\pi R$. Then the condition $A_m \ll A_c$ reduces to $1-T/T_c \gg [20I\sqrt{3}\xi_0/R]^2$, where $I = \Phi/\phi_0$. For $R = 13\text{mm}$, $\xi_0 = 9\text{ nm}$ (extracted from $\xi(25.8\text{K})=25\text{ nm}$ on top of p.7) and the field in the coil = 1 Oe corresponding to the number of flux quanta $I=10^4$ (see my first report), the London theory works if $1-T/T_c \gg 0.06$. This rough estimate shows that the London model is hardly applicable at $29.5 < T < 30\text{ K}$ where the stiffnessometer shows zero signal. If $A(x,y,z)$ reaches A_c somewhere at the surface, the Meissner current flow becomes absolutely unstable against penetration of vortices. This instability can be facilitated by the electron mass anisotropy and penetration of vortices at materials defects at the surface at $A < A_c$. As a result, the temperature region where the London theory breaks down further widens as compared to the above estimate. In this crucial region of T the proper analysis of the stiffnessometer data should be done using the anisotropic Ginzburg-Landau equations.
3. To convince the reader that their analysis is adequate the authors could compare their calculated current maps with the applicability conditions of the London model $J \ll J_d$, taking into account the crystal anisotropy and realistic materials and geometrical parameters of their rings at $29.5 < T < 30\text{ K}$. Unfortunately, the results shown in Fig. 10 do not convince me that the London model is applicable at $29.5 < T < 30\text{ K}$ because the data were taken at 25.8K which is not very close to T_c .

In conclusion, I re-iterate that the analysis of the stiffnessometer data based on the London model is inadequate at T close to T_c where the interplane phase coherence is likely destroyed by penetration of Josephson vortices along the ab planes. For this reason, I am not sure that this experiment can actually measure the London penetration depths and the superfluid density close to T_c , so a discussion of different interpretations of this experiment would be appropriate.

In what follows we address each one of referee 4 comments individually:

Referee> In the revised version the authors added a much-needed inset in Fig. 3 which shows the conditions under which the London model in the c-ring is applicable at 29.2K. From Fig. 3 and Fig. 10 in Suppl. B of the previous version I can infer the critical currents (above which the London model breaks down): $I_c = 3 \text{ mA}$ at 25.8K and $I_c = 1 \text{ mA}$ at 29.92K. Thus, for all green dots in Fig 3a taken at $I=3\text{mA}$ the London analysis certainly breaks down because 25.8K is way outside the T-scale in Fig. 3a.

Reply> The referee is correct. Indeed, the data used to extract λ was done using measurements at $I=1.0\text{mA}$.

Referee> The magenta dots to the right of the arrow on the decreasing part of $\Delta V_R(T)$ for the c-ring are also beyond the applicability of the London model.

Reply> In the new version we mark in Fig. 3a for both the a-ring and c-ring the temperature up to which $I < I_c$. We clearly state on page 3 second paragraph that the London equation is valid only up to the temperatures where $I < I_c$.

Referee> The problem becomes far more serious for the a-ring for which I_c is reduced by anisotropy, so $I = 0.5, 1$ and 2 mA are likely above $I_c(T)$ at for most of the datapoints on the decreasing part of $\Delta V(T)$. Thus, the majority of the datapoints in Fig. 3a were apparently taken at currents for which the London applicability condition $I \ll I_c$ is not satisfied. These experimental data just reinforce the main point of my previous review.

Reply> This is an excellent point by the referee. To remove any doubt, we added to Fig. 3a inset a measurement of the a-ring critical current. Using this measurement, we indicated in Fig. 3a until which point $I < I_c$, namely, until which point the London equation is valid. Again, we clearly state on page 3 second paragraph that the London equation is valid only up to a certain temperature.

Referee> The applicability of the London model was briefly discussed in the text, but surprisingly the crucial condition $A \ll A_c(T)$ (see my report) was not even mentioned either in the bottom left paragraph on p. 2 or in Suppl. B3. The authors are aware of this condition, yet they keep insisting in the abstract and in the text that the London model is applicable all the way to $T \rightarrow T_c$ despite the fact that $A_c(T_c) = 0$, so the London model always breaks down at $T \rightarrow T_c$.

Reply> We are somewhat puzzled by this comment since in the abstract we wrote "The method is based on the London equation...it remains so as long as A is smaller than some critical value". In the new version we emphasize this point further by rewriting the paragraph on the bottom of page 2 left side. Now it says:

"There is a range of applied currents for which using the London equation alone rather than the full Ginzburg-Landau theory or Pippard relation is justified because...

This value of I does not change as A is turned on, and London's equation holds as long as $A < A_c(T)$, where $A_c(T)$ is a critical value of the vector potential which tends to zero as $T \rightarrow T_c$."

Referee> The problem is not to show that $\Delta V(I)$ is linear at $I < I_c$ (which is always the case for small I) but to actually measure $\lambda(T)$ at $I < I_c$. It is this condition $I \ll I_c(T)$ which is apparently not satisfied for the majority of the datapoints on the decreasing parts of $\Delta V(T)$ in Fig. 3. Moreover, it is unclear whether most of the datapoints shown in the inset of Fig. 3b were obtained at $I > I_c(T)$ (no information was given), and if so, how relevant they are to the actual values of $\lambda(T)$. As T approaches T_c , it becomes increasingly difficult to ensure $I < I_c(T)$ experimentally, but this problem should be acknowledged and

discussed because $\lambda(T)$ extracted at $I > I_c(T)$ can give a misleading picture of superconductivity in cuprates. One cannot just ignore the fact that the extraction of $\lambda(T)$ very close to T_c where $I_c(T)$ becomes very small requires numerical simulations of the anisotropic Ginzburg-Landau equations.

If the authors want to use the London model for a qualitative analysis of their experimental data, they should spell out all key limitations of their method and state that the datapoints outside the applicability of the London model can only give qualitative trends but hardly the actual values of $\lambda(T)$. In my opinion this straightforward approach would be acceptable for the publication of this first experimental work showing how the Stiffnessometer can be used to extract valuable physical characteristics of the cuprates.

Reply> This is an excellent point by the referee. To remove any doubt, we added to Fig. 3b an indication until which point $I < I_c$, namely, until which point the London equation is valid. We write in the text on the first paragraph of page 4 that “The extraction of λ_{ab} is from our $I=1.0\text{mA}$ data and is valid only as long as $I < I_c$. This point is marked in the inset. Beyond that point the calculated values only give qualitative trend but hardly the actual values of $\lambda_{ab}(T)$ ”. We do the same when extracting $\lambda_c(T)$. On page 5 first paragraph we added “The extraction of λ_c is from our $I=1.0\text{mA}$ measurement and is again valid only as long as $I < I_c$. This point is marked in the inset. As stated before, beyond that point the extraction only gives qualitative trends of $\lambda_c(T)$ ”. An appropriate adjustment was done to the caption of Fig. 3 as well.